# The Control of Cell Expansion, Cell Division, and Vascular Development by Brassinosteroids: A Historical Perspective

**DOI:** 10.3390/ijms21051743

**Published:** 2020-03-04

**Authors:** Man-Ho Oh, Saxon H. Honey, Frans E. Tax

**Affiliations:** 1Department of Biological Sciences, College of Biological Sciences and Biotechnology, Chungnam National University, Daejeon 34134, Korea; manhooh@cnu.ac.kr; 2Department of Molecular and Cellular Biology, University of Arizona, Tucson, AZ 85721, USA; honey@email.arizona.edu

**Keywords:** brassinosteroids, cell expansion, cell elongation, cell division, vascular differentiation, *Arabidopsis thaliana*

## Abstract

Steroid hormones are important signaling molecules in plants and animals. The plant steroid hormone brassinosteroids were first isolated and characterized in the 1970s and have been studied since then for their functions in plant growth. Treatment of plants or plant cells with brassinosteroids revealed they play important roles during diverse developmental processes, including control of cell expansion, cell division, and vascular differentiation. Molecular genetic studies, primarily in *Arabidopsis thaliana*, but increasingly in many other plants, have identified many genes involved in brassinosteroid biosynthesis and responses. Here we review the roles of brassinosteroids in cell expansion, cell division, and vascular differentiation, comparing the early physiological studies with more recent results of the analysis of mutants in brassinosteroid biosynthesis and signaling genes. A few representative examples of other molecular pathways that share developmental roles with brassinosteroids are described, including pathways that share functional overlap or response components with the brassinosteroid pathway. We conclude by briefly discussing the origin and conservation of brassinosteroid signaling.

## 1. Introduction

Brassinosteroids (BRs) are an essential plant hormone and are now considered at the same level as the classic plant hormones auxin, cytokinin, ethylene, gibberellic acid, and abscisic acid. This is due to the important roles of BRs in regulating developmental processes also controlled by the classic plant hormones such as cell expansion, cell division, and differentiation, and because of the well-characterized biosynthesis and signaling pathways of BRs. In this review, we take a historical perspective and begin by describing the initial chemical characterization of plant steroid hormones and their bioactivity in different plant-based assays for cell elongation, cell division, and vascular development. Next, we describe the identification of the components of the signaling pathway that perceives BRs, which is one of the best-characterized pathways in plants. We discuss the genetic evidence for the roles of the BR pathway components in the developmental processes of cell elongation, in the control of cell division, and in vascular development. Other influences or pathways that participate in cell elongation are also discussed, including potential integration points with some signaling pathways of the classic plant hormones. Finally, we summarize the evidence for the conservation of BR signaling throughout the plant kingdom. Below is a timeline (Figure 1) that delineates some of the key breakthroughs in BR research.

## 2. Brassinosteroids-Structure and Functions

### 2.1. The Isolation of Brassinosteroids, Plant Steroid Hormones

The discovery of brassins, a naturally occurring group of plant-growth-promoting substances purified from pollen, was first reported by a research group at the United States Department of Agriculture [1]. To obtain enough crude extract, about 225 kg of bee-collected *Brassica napus* pollen was processed via a pilot plant-size solvent extraction procedure and partially purified [12]. Following this purification, crystals were obtained and subjected to X-ray crystallography to determine their structure [2]. The isolation and structural elucidation of brassin [2], a novel polyhydroxysteroidal lactone with strong plant growth-promoting activity, was the beginning of intensive study of a new class of phytohormones, termed brassinosteroids (BRs). The most biologically active plant-growth-promoting steroid is brassinolide (BL, C28H48O6, MW = 480, see Figure 2). The structures of BL, 24-Epibrassinolide (another naturally occurring BR), and progesterone, for comparison, are shown in Figure 2. 24-Epibrassinolide was one of the first BRs to be synthesized efficiently in the lab [13].

### 2.2. Biological Activities of Brassinosteroids

Brassinosteroids were initially characterized for their growth-promoting effects in plants. Plant growth could theoretically be stimulated by changes in cell elongation (cells expanding in one dimension to elongate cells and organs) or expansion (cells expanding in all three dimensions), by cell division, or by changes in differentiation. In this section, we will summarize the results of BR assays that measure cell expansion, cell division, and vascular differentiation (as an example of differentiation). BRs are also implicated in both biotic and abiotic stress: these have been reviewed elsewhere and are not discussed here.

Three assays that were developed to measure the activity of BRs are shown in Figure 3: the lamina joint test (LJT) in rice, the hypocotyl elongation assay in Arabidopsis, and the second internode assay in beans. The Lamina Joint Test (LJT) (Figure 3A) was initially developed in rice as a standard growth assay for measuring auxin responses [14]. Briefly, developing rice leaves are composed of two areas separated by the joint. The distal part of the emerging leaf blade is called the lamina, and the region of the leaf that surrounds the stem is called the sheath (Figure 3A). If there is significant growth in the lamina, the leaves are more susceptible to gravity and grow at an increased angle to the main stem. If growth is repressed in the lamina, the leaves are more erect, with a smaller angle to the stem. When there is growth of the lamina, the cell length on the adaxial side (toward the stem) of the lamina is four or five times as long as the cells on the abaxial side (away from the stem) [14,15]. The LJT was also used as a microquantitative analytical assay for BL, and investigators examined the sensitivity of compounds from various cultivars and species to identify bioactive compounds. These assays showed that BL activity levels are ten-thousand-fold higher than the activity of auxin (IAA) [16]. Subsequently, new analogs of BL were isolated and identified from many kinds of plants using this bioassay [17]. For example, the beginning of BR research in Japan was initiated with research on Distylium (winter hazel, related to witch hazel) factors, which were discovered in rapidly expanding galls induced by aphids in *Distylium racemosum* leaves. These factors, when assayed using the LJT and characterized biochemically, were found to be BRs, as expected [18].

The second assay that is commonly used in assays for plant growth is the elongation of the hypocotyl (Figure 3B). In light-grown plants, the hypocotyl is the short region of the stem between the root (initiated at the proximal end of the root where the vasculature changes into the pattern typical of the stem) and the stems of the cotyledons. In dark-grown plants, the hypocotyl elongates significantly, allowing seedlings to orient (via root gravitropism) and grow quickly towards the light present at the surface of the soil. The hypocotyl growth assay was used to identify mutants in the response of plants to different types of light [19]. Briefly, plants were screened in the dark or under specific wavelengths of light, and those that failed to elongate were studied. These screens allowed for the identification of mutants in genes necessary for responses to red and blue light. For studies examining mutants failing to respond to growth in the dark, identified mutants included those in BR synthesis and BR response genes because the cells in the hypocotyl were unable to elongate. For plants with small seeds and seedlings, such as Arabidopsis, the hypocotyl elongation assay can be performed on Petri dishes containing agar as it provides a transparent surface convenient for tracking root and hypocotyl growth. Changes in the length of the hypocotyls after treatment can easily be measured by eye or by using a scanner and computer analysis. Analysis of cells in the hypocotyls of BR synthesis mutants revealed that cell expansion was the primary defect in the mutants, and this phenotype could be complemented by adding BR biosynthetic intermediates after the blocked step in the mutant [20].

The third assay we will discuss is the second internode assay, which was commonly used in studying beans. For this assay, plants are grown for sufficient time for two or three pairs of leaves to develop after the opening of the cotyledons. The substances to be tested are then applied to the second internode—the section of the stem between where the first and second true leaves emerge. The increase in elongation of this specific section of the stem is then measured (Figure 3C). The second internode assay was used [1,2,21] to initially characterize the action of brassins. Increases in both cell elongation and cell division were noted, so at the time that brassinosteroids were first characterized, it was already clear that both cell size or division could be responsible for the increase in plant growth. However, many investigators did not clearly distinguish between cell expansion and cell division or determine if both were affected to the same degree in their assays.

### 2.3. Brassinosteroids and Cell Expansion

In general, the main growth-promoting role of BRs was considered to be cell expansion or elongation [22]. Early studies in pollen, which are single cells, indicated pollen tube growth, through cell expansion, was stimulated [3]. Cell growth has also been studied extensively in whole plants or plant organs by looking at the dynamics of auxin, gibberellic acid, and other plant hormones, and this led to the acid growth hypothesis [23,24]. This model proposed that auxin, or other growth-promoting substances, led to the activation of plasma membrane ATPases, which expel H+ out of cells into the apoplast, the region between the cell membrane and cell wall. The increased acidity in the apoplast, due to additional protons, was thought to cause the carbohydrates and proteins that make up the cell wall to loosen their interactions, allowing the cell wall to expand and the cell to become longer or larger.

Investigators also reasoned that additional changes to the cell wall might be required to accommodate cell expansion. For example, one group [25] reported that hypocotyl elongation of pak choi (*Brassica rapa chinensis*) was stimulated by BR. Using pressure-block experiments, they concluded that BR stimulates growth in pak choi stems by accelerating the biochemical processes that cause wall relaxation, without inducing a large change in wall mechanical properties. This prompted attempts to isolate genes that might respond to BR treatment: Zurek et al. [25] constructed a cDNA library from BR-treated soybean epicotyls and used differential hybridization to isolate a cDNA (pBRU1) corresponding to a transcript whose abundance is increased by BR treatment. The sequence had significant homology to a xyloglucan endotransglucosylase (XET) localized in the cell walls of nasturtium [26]. XETs are proposed to act by cleaving xyloglucans in cell walls and then restitching them together, consistent with a role in the changes that might accompany cell wall expansion. RNase protection studies showed that BRU1 expression is highest in stem tissue. The elevated levels of BRU1 transcripts in elongating tissue and the homology with XET suggest a possible role for the BRU1 protein in BR-stimulated elongation [4]. 

As mentioned previously, the early thinking was that brassinosteroids were associated mainly with cell elongation. Using the hypocotyl elongation screen described above (Figure 3B) [5,19], mutants were identified that did not elongate in the dark. When these mutants were allowed to grow in soil, they grew with an extreme dwarf phenotype (Figure 1). Some of these mutants with dwarf phenotypes were able to be partially rescued by BR treatment, and it was reasoned that these affected genes might encode biosynthetic enzymes. Other mutants with similar dwarf phenotypes that were insensitive to BR treatment were classified as potential BR signaling genes [4]. Through this analysis, more than ten steps were identified in BR biosynthesis and several kinases (BRI1 and BIN2) were identified; we now know these are key steps in BR signaling (see below).

When a brassinosteroid deficient mutant, CONSTITUTIVE PHOTOMORPHOGENESIS AND DWARF *(cpd)* was analyzed, epidermal cell lengths in the hypocotyl were reduced approximately five-fold. However, not all cell types or organs were affected by the same magnitude; the length of root epidermal cells was reduced by 20–50% in these same mutants [27]. After growth in soil for five to eight weeks, brassinosteroid dwarf mutants are often 10- to 20-fold smaller than wildtype control plants of a similar age [27,28]. Cell elongation and cell expansion in BR deficient mutants were analyzed extensively in another study [29], where it was found that cells were shorter in length (50–80% less), width (50–80% less), but thickness was not changed significantly (10–20% thicker). These results indicated that 1) cell expansion in two dimensions was affected in leaf cells, and that 2) other differences (potentially including cell division) significantly impact the growth of these BR-deficient mutants.

### 2.4. Brassinosteroids and Cell Division

Increasing the number of cells in a tissue or organ is important for growth, and there is substantial evidence that BRs control cell division as well. The results of second-internode bioassays in beans have suggested that BRs affect cell division as well as elongation [30]. Cell division is stimulated by BRs (in the presence of auxin and cytokinin), in cultured parenchyma cells of *Helianthus tuberosus* (Jerusalem artichoke) [31], and in protoplasts of Chinese cabbage and petunia [32,33]. Preliminary results have also shown that BR affects the kinetics of the cell cycle in synchronized cell cultures of tobacco and also regulates the expression of genes associated with the S-phase, including H2B and High Mobility Group-1 protein [34]. Moreover, the expression of CDC2b cyclin-dependent kinase is upregulated by BRs in the dark but remains unaffected in the presence of light [34]. Treatment of *det2* (DE-ETIOLATED-2, a mutant isolated in the hypocotyl growth screens described in Section 2.2 above) cell suspension cultures with 24-epibrassinolide increases transcript levels of the gene encoding cyclinD3 (CycD3), a protein involved in the regulation of G1/S transition in the cell cycle [35]. In the studies of cell expansion and cell division in *cpd* mutants (Section 3), there were fewer mesophyll cells in leaves, but this was not quantitated [27]; this result would predict that brassinosteroids positively regulate cell division. This idea was substantiated for leaf cells when Nakaya et al. [28] analyzed the cell number of leaves from DWARF1 (*dwf1*) and *det2*. They found that the palisade and mesophyll cell in leaves, which are typically shorter but not narrower in these dwarf mutants, had 50–60% fewer cells when longitudinal sections of the leaf were taken, but transverse sections did not consistently have fewer cells. As an additional result emphasizing the importance of BRs in cell division, an Arabidopsis BR biosynthetic mutant, DWARF-7-1 (*dwf7-1*), showed diverse characteristics attributable to slower cell division rates including callus growth and shoot induction relative to a wild-type control. Expression levels of the genes involved in cell division and shoot induction, such as PROLIFERATING CELL NUCLEAR ANTIGEN2 (PCNA2) and ENHANCER OF SHOOT REGENERATION2 (ESR2), were also lower in *dwf7-1* as compared to wild type [36]. Zhiponova et al. showed that the reduced size of *cpd* leaf blades is a result of a decrease in cell size and number [37]. Investigation of cell cycle markers in leaves in *cpd* mutants revealed accumulation of mitotic proteins, independent of transcription. This correlated with an increase in cyclin-dependent kinase activity, suggesting a role for BRs in control of mitosis.

One lesson from these studies addressing BR regulation of cell expansion or cell division is that there are many differences among cell types, but both processes are often affected to some degree. One excellent example is a recent description of the cell growth during the development of the lamina joint in rice (Section 2.3). It was known that the cells on the adaxial side (toward the stem) of the lamina elongated more in wild-type strains; subsequent experiments identified a cyclin as a transcriptional target of BRs in the lamina [38]. Another interesting finding was that cell division also plays an important role in the angle of the joint; when BRs are present, cell division in the abaxial side (facing away from the stem) is inhibited. This is in addition to the elongation of cells on the adaxial side. When BRs are not present, or not sensed in a BRI1 mutant, cell division in the abaxial side is increased, further reinforcing the erectness of the leaves. Therefore, in the lamina of rice leaves, cell expansion and cell division both play important roles.

### 2.5. Cell Culture Studies of Vascular Development

The vascular system is comprised of bundles of connected tubes that transport materials throughout the roots, stems, leaves, and flowers of plants. This vascular system is an evolutionary innovation that has contributed greatly to the success of the land plants. Without these specialized conducting and connective tissues, plants would be limited to a size of approximately an inch based on the limitations of the free diffusion of water through tissues [39].

The three major specialized cell types in the vasculature include the xylem, which transports water and dissolved nutrients from the roots to the shoots for transpiration; phloem, which is specialized for the transport of sugars from photosynthetic tissues to sink tissues that require the sugars; and the stem cells, which are also known as procambium or cambium [40]. Cells in the xylem and phloem undergo specialized differentiation pathways that modify their cell walls, cause the cells to undergo cell death, and dissolve the apical and basal ends of the cells—enabling cells to form continuous tubes known as tracheids or vessel elements (xylem) or sieve elements (phloem). These interconnected tubes allow movement of water, solutes, and sugars as needed. Both xylem and phloem also transport additional substances such as hormones, RNAs, proteins, and small peptides. The cell wall modifications of vessel and sieve elements also provide structural support for plants and are readily visible under a light microscope.

When plants grow, cell divisions create undifferentiated cells that are recruited for distinct cell fates. There is a clearly established role for the local and directional effects of the plant hormone auxin (IAA) for recruiting cells during the formation of provascular cells from undifferentiated tissue [41,42]. The first notable difference is in leaf cells that will become provascular cells: they stain more densely than their neighbors—most likely because they are highly vacuolated [43]. Provascular cells are also long and narrow and will divide longitudinally, creating elongated cell clusters at times when most of the leaf cells are dividing in the direction of new leaf outgrowth.

The spatial order of vascular stem cells, xylem, and phloem can vary between leaves, different parts of the stem, and roots, but there are several important commonalities. The stem cells, which are always located between the xylem and phloem, divide asymmetrically, and the fate of each daughter cell depends on whether it divides towards the xylem or the phloem. The coordination of vascular growth is complex: vascular meristems must also coordinate their cell divisions such that rows of cells remain end-to-end to form conductive elements for efficient transport.

### 2.6. Role for Brassinosteroids in Vascular Development Established Using Cell Culture Systems

Because of the importance of the vascular system in plants, several types of in vitro cell culture systems were developed to determine the developmental mechanisms necessary for producing specific vascular cell types. One such culture system was developed from tuber explants from *Helianthus tuberosus*—the Jerusalem artichoke. As with many in vitro culture systems, addition of the plant hormones auxin and cytokinin, induced differentiation of tuber explants into tracheary elements (TEs)—the differentiated cells of the xylem. The addition of nanomolar concentrations of BL increased the rate of TE formation (identified through helical patterns of cell wall differentiation, normally requiring 3–4 days) ten-fold after 24 h. In addition, after three days, there were both more TEs as well as a greater number of total cells, indicating that BRs promote cell division in this culture system [44]. These results suggest that one of the roles of BRs is to promote xylem differentiation.

Similar results were obtained through the *Zinnia elegans* culture system, in which single mesophyll cells are induced to differentiate into TEs. The general cytochrome P450 inhibitor, uniconazole, was found to inhibit differentiation of mesophyll cells into TEs but did not affect cell division [45]. Further addition of BL countered the inhibition of uniconazole in this system, supporting the idea that BL synthesis is the key target of uniconazole in TE differentiation. Similar experiments were conducted with gene expression as the output of differentiation rather than the appearance of TEs, and these experiments confirmed that uniconazole inhibits TE formation and that BL treatment can bypass the effects of this inhibitor [46].

## 3. Brassinosteroid Signaling

The BR signaling pathway is one of the best-understood plant signaling pathways. As described in Figure 4, BRs are perceived at the plasma membrane, where they directly bind the extracellular leucine-rich repeat (LRR) domain of a receptor-like kinase, BRASSINOSTEROID INSENSITIVE1 (BRI1) [47,48,49]. In the absence of BRs, or when low levels of BRs are present, BRI1 KINASE INHIBITOR 1 (BKI1) interacts with BRI1 and inhibits its function [50]. Consequently, a member of the cytoplasmic GLYCOGEN SYNTHASE KINASE-3/SHAGGY (GSK3/SGG) kinase family member BIN2 (BRASSINOSTEROID INSENSITIVE 2) phosphorylates the transcription factors BZR1 and BES1/BZR2 (bri1 EMS SUPPRESSOR1/BRASSINAZOLE RESISTANT1) [51,52], keeping these transcription factors inactive.

After perception of BRs by BRI1, BKI1 is phosphorylated on a Tyrosine residue and disassociates from BRI1 [53,54], which leads to the association of BRI1 with its coreceptor BRI1 ASSOCIATED KINASE1 (BAK1) [6,55]. Autophosphorylation and transphosphorylation between BRI1 and BAK1 then lead to the activation of the BRI1 kinase [56]. Activated BRI1 phosphorylates the Receptor-Like Cytoplasmic Kinases (RLCKs), BSK1 and CDG1, which is followed by the phosphorylation and activation of the phosphatase BSU1 (BRI1 SUPPRESSOR 1) which dephosphorylates BIN2. Subsequently, PROTEIN PHOSPHATASE 2A (PP2A) dephosphorylates BZR1 and BES1/BZR2 [57], which then become transcriptionally active to regulate many different genes in plant growth and development [58,59,60]

There are three major themes in receptor kinase signaling that were all uncovered during studies of Brassinosteroid signaling. First, the phosphorylation and oligomerization of BRI1 and BAK1 are early events in BR signaling, and phosphorylation and/or oligomerization have been a common theme in RLK-mediated signaling. Additionally, since the discovery that BAK1 interacts with BRI1, BAK1 has been found to interact with many other LRR-RLKs [61,62,63,64]. Second, the use of phospho-specific antibodies demonstrated that BRI1 was a dual-specificity kinase capable of phosphorylation on both Serine/Threonine and Tyrosine residues, which was the first demonstration of the importance of Tyrosine phosphorylation by RLKs in plants [65]. Third, while a simplistic pathway is shown in Figure 4, it should be noted that there are multiple orthologs involved in each step. For example, BRL1 and BRL3, close homologs of BRI, can bind brassinosteroids. There are multiple glycogen synthase kinases in Arabidopsis, including BIN2 [66], and multiple BSKs related to BSK1 [67].

There are still many gaps in the BR pathway and downstream responses, such as cell division and cell expansion. A recent study of the Arabidopsis H+ ATPase 2 (AHA2) showed that AHA2 activity is regulated by phosphorylation of a key Threonine residue [67]. Phosphorylation is reduced in a *bri1* mutant, and not BR dependent in a *bin2* mutant, suggesting that the BR pathway directly phosphorylates AHA2. However, the steps between BIN2 and AHA2 are not clear, and there are at least two places where a 14-3-3 protein could act. These downstream steps need to be characterized and the ambiguities in the pathway resolved. In the analysis of cell division in the rice lamina joint [38], BES1 binds to the promoter of the cyclin to inhibit its expression, and BIN2 phosphorylates the cyclin directly to activate DNA replication. In this example, two components of the canonical BR signaling pathway act to regulate cell division depending on the presence or absence of BRs.

## 4. Role for Brassinosteroids in Vascular Development from Arabidopsis Mutants

These in vitro results are consistent with the hypothesis that BRs promote xylem differentiation, but the defined molecular pathways are difficult to unmask in these culture systems. Further insight into the roles of BRs in vascular development came from the initial analysis of the mutants in genes required for BR biosynthesis, described earlier. Analysis of the stems of *cpd* mutants, encoding a BR biosynthetic enzyme, revealed that there were more phloem cells in the vascular bundles compared to wild-type [68]. Later studies of mutants in the DWF7 gene, which is a more upstream biosynthetic enzyme, found a reduced number of xylem cells and also a reduction in the number of vascular bundles within stems, and these vascular bundles were also sometimes fused together [69].

As expected, mutants in the major brassinosteroid receptor, BRI1, have similar defects in the vasculature. Analysis of the vascular cell types in a weak BRI1 mutant, *bri1-5*, revealed that there were also fewer xylem cells and more phloem [7]. However, several other receptors related to BRI1 might also be expected to function in vascular development since they also bind steroids. Two receptors, BRL1 and BRL3, are functional BR receptors based on two criteria: (1) BRL1 and BRL3 are able to bind 3H Brassinolide in an in vivo assay, and (2) BRL1 and BRL3 coding sequences could replace BRI1 sequences and fully rescue a weak *bri1* mutant. BRL1 and BRL3 would be predicted to function in vascular development as both receptors were expressed in vascular cells in roots and leaves [7].

To test whether BRL1 and BRL3 also function in vascular development, single mutants in each receptor were analyzed for vascular defects. When mutants were isolated in the standard Columbia accession, both *brl1* and *bri3* single mutants did not appear to have mutant phenotypes, and the *brl1* and *brl3* mutants did not enhance a *bri1* null mutant. However, when all three BR receptor mutants were combined in one strain (*bri1-101*; *brl1-2*; *brl3*), xylem differentiation phenotypes were more extreme than in the *bri1* or *brl1* single mutants: xylem differentiation appeared to be delayed, with fewer xylem cells and fibers. In addition, the phloem appeared to differentiate abnormally. These studies indicate that BRs may play an important role not only in xylem differentiation but also in phloem differentiation. The phloem-specific expression of BRL3 was found to depend on BRs in a BES1-dependent manner, and BES1 was found to bind directly to the promoter of BRL3 [70]. BRL1 was not found to be regulated by BRs and BES1, so BRL1 and BRL3 likely signal through distinct pathways, and also not through the precise pathway shown in Figure 4.

The hypothesis that BRs function in phloem differentiation was confirmed by more recent studies from [71], who analyzed the differentiation of the protophloem in *bri1* and *bri1*; *brl1*; *brl3* triple mutants and found that gaps in normally continuous cells of the phloem could be identified in just the triple mutants. Interestingly, this phenotype could not be rescued by treatment with the GSK and BIN2 inhibitor bikinin. This suggests that this phenotype of *bri1*; *brl1*; *brl3* plants does not utilize the canonical BR signaling pathway and may signal through one of the bikinin-resistant GSKs in Arabidopsis [66] or not signal through a GSK at all (Figure 4).

Analysis of additional BR biosynthesis and signaling mutants produced more evidence that BRs regulate the number of vascular bundles. In addition to confirming that mutations in BR biosynthesis or signaling components reduce the number of vascular bundles, Ibanes et al. [71] found that increased BR synthesis, signaling, or mutations that activate BR signaling increase the number of vascular bundles. Analysis of cell numbers within and between vascular bundles in these mutants or transgenic lines indicated that there were more cells in the plants from the increased BR signaling group, further emphasizing a role for BRs in cell division of the cells that will form the vasculature. How BRs sequentially regulate cell division and differentiation in stem vascular cells is not yet understood, but there may be differential and hierarchical roles of BRI1, BRL1 and BRL3.

The role for VH1, or BRL2, one of the three LRR RLKS that are most closely related to BRI1 is less clear. VASCULAR HIGHWAY-1 (VH1) was originally identified as an enhancer trap line that was specifically expressed in the provascular cells, the stage in vascular development before vascularization has taken place. Overexpression of VH1 caused fewer veins to develop, and cells differentiated prematurely—or additional cell divisions were detected—whereas plants homozygous for a loss-of-function *vh1* mutant had reduced phloem transport and underwent premature senescence [72]. VH1 was not able to bind BRs or replace BRI1 in functional assays, so it is currently not thought to be a BR receptor. From these results, a role for VH1 in vascular differentiation is clear, but how this phenotype relates to the phenotype of BRI1, BRL1, and BRL3 mutants has not been established.

## 5. Other Receptors for Cell Elongation-FERONIA

Another receptor kinase regulating cell elongation is FERONIA, a receptor with two extracellular malectin domains that can interact with cell wall components such as pectin. FERONIA was initially discovered in a pollen tube mutant screen; *fer* mutant plants have severely impaired fertility due to the pollen tube failing to arrest and consequently continuing to grow and elongate within the gametophyte [73]. FERONIA was later identified as an important modulator for cell growth and hormone response (e.g., synthesis of auxin and response to ethylene; see below); it was subsequently named after the Etruscan fertility goddess.

One identified ligand for FERONIA is RALF (Rapid ALkalinization Factor)—a 5 kDa secreted peptide. The RALF protein family is highly conserved and predates the division of plants into monocots and dicots; although the family contains many genes, only a few have been characterized [74]. The name *rapid alkalinization factor* comes from its original discovery: purified from tobacco, the unidentified peptide caused rapid alkalinization of culture medium [75] and activation of a mitogen-activated kinase. Upon overexpressing RALFs in tomatoes and Arabidopsis, RALF arrested root growth and development. FERONIA, when bound to RALF, directly phosphorylates AHA2, inhibiting proton transport, consequently inhibiting cell growth [76] in line with the acid growth theory. Later experiments showed that RALF is also involved in stress responses; in the event of a pathogenic attack, RALF signaling induces an increase in cell pH, resulting in less pathogen growth.

In addition to suppressing cell expansion from the plasma membrane, FERONIA is also involved in several intracellular processes related to cell growth, including expansion of the large vacuole found in plant cells, and modulation of BR responsiveness. FERONIA regulates vacuolar size through interactions with Leucine-Rich Repeat Extensin (LRX) proteins, as seen in Figure 5. The LRX proteins are located between the plasma membrane and the cell wall, and the EXT domain interacts with the cell wall. The LRR domain binds to FER and acts as a conduit between FER and the cell wall, communicating the constraint of the cell wall [77]. Interaction between FERONIA and LRX causes FERONIA to become phosphorylated and suppress vacuolar expansion through a mechanism yet to be discovered. As the largest plant organelles, vacuoles contribute substantially to cellular expansion; due to their ability to fill space within the cell, vacuolar expansion allows for swift elongation by reducing the cell’s need to produce extraneous cytosolic content.

FERONIA’s role in BR responsiveness was demonstrated by Deslauriers et al. [78], who showed this role might be indirect and dependent on ethylene signaling. Plants with RNAi knockdown of FERONIA showed a normal response to BRs grown in light, whereas a full loss of function knockout resulted in extreme hypersensitivity to 24-epibrassinolide (EBL) [78]. This suggests that in the absence of FER, the ATPase is more active, allowing BRs to amplify the cell elongation response.

FER and BRI1 appear to have similar downstream enzymes such as AHA2. It will be interesting to identify other potential areas of crosstalk between these receptors that both control cell elongation. Potential mechanisms for crosstalk could include those through co-receptors such as BAK1, through cytosolic kinases, or through 14-3-3 adapter proteins. It will be interesting to see if these mechanisms are used or if new proteins in their pathway are shared.

## 6. Connections to Other Hormones and Processes Involved in Cell Expansion, Cell Division and Differentiation

In addition to determining how the BR signaling pathway is used in different cellular contexts and cell types, the next few years will also see more emphasis on understanding how a single process such as cell elongation is controlled by multiple hormones and by varying environmental conditions. Analysis of microarrays did not support the idea that multiple hormones operated through overlapping transcriptional targets [79]; rather, hormones operated through interconnected transcriptional mechanisms. BR and auxin are both known to affect cell expansion, and their signaling pathways have been shown to directly interact through the phosphorylation and subsequent inhibition of the transcription factor AUXIN RESPONSE FACTOR-2 (ARF2) by BIN2 [80], suggesting that BR would promote the transcriptional activity of ARF2, which is a repressor belonging to the ARF family of transcription factors that consists of both activators and repressors. Because auxin and BRs regulate common target genes, removal of ARF2 activity is expected to mimic BR response, as demonstrated by the physiological analysis of *arf2* mutants [80].

Environmental factors can also interact with hormone signaling and transcriptional responses. A complex molecular mechanism was identified, linking BR, gibberellins, temperature, and light-mediated signaling [81,82,83] through different transcription factors. For example, BZR1 forms heterodimers with the light- and temperature-regulated transcription factor PHYTOCHROME-INTERACTING FACTOR 4 (PIF4) and regulates the expression of genes necessary for cell elongation. Under low gibberellin conditions, DELLA protein accumulates and binds BZR1 and PIF4, therefore inhibiting their activity and subsequently the induction of secondary transcription factors, specific PREs [84]. Hence, cell elongation is coordinated by various internal and environmental cues. The BZR1–PIF–DELLA module can account for cell elongation in hypocotyls and leaves; whether it has the same effect in other organs, like roots, remains an open question. As with BZR1, BES1/BZR2 binds PIF4 and DELLA, but has also been shown to interact with other transcription regulators involved in cell elongation. These results show the many connections between BR and other hormones, including through BIN2, BES1, and BZR1, and hint at many additional links that remain to be discovered.

## 7. Conservation of Brassinosteroid Signaling in Plants

BRs are found throughout the plant kingdom, with some evidence supporting the existence of BRs in all vascular plants. Selaginella, an ancestral plant, appears to have cytochrome P450s that resemble some of the enzymes that perform biosynthetic steps in BR biosynthesis, as well as signaling components such as BIN2 and BZR1 [85]. Treatment of plants with BR or a BR inhibitor have opposing effects on growth, consistent with a role for BRs in Selaginella. However, no RLK with a BR-binding island domain like that found in BR11 could be found, suggesting that BR perception is through a different motif. Phylogenetic analysis also found island domains containing LRR RLKs only in the angiosperms and gymnosperms [86,87]. As more research is conducted in the next decades, it will be of interest to see which signaling components are conserved in various model systems and crop plants, and how plants have evolved different mechanisms to manage growth under specific environmental conditions.

## Figures and Tables

**Figure 1 ijms-21-01743-f001:**
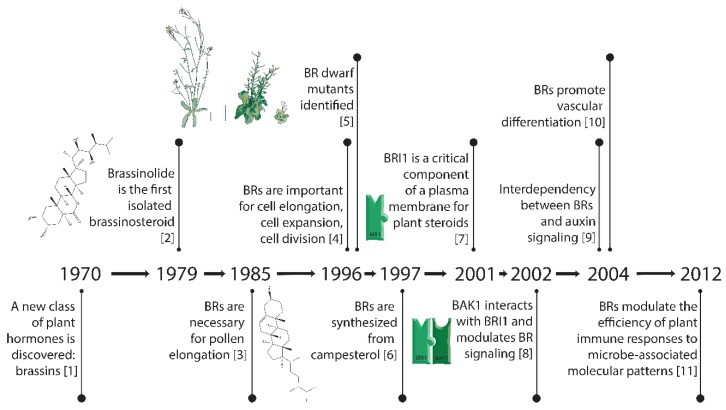
Key breakthroughs in the timeline of research on brassinosteroids [1,2,3,4,5,6,7,8,9,10,11].

**Figure 2 ijms-21-01743-f002:**
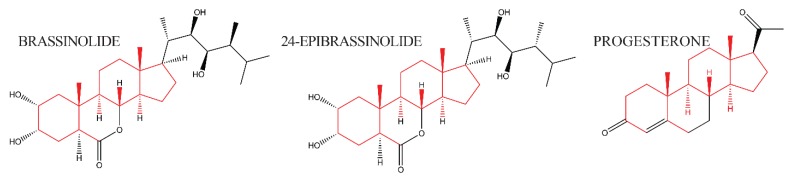
The structure of brassinolide and 24-epibrassinolide, plant steroid hormones related to animal steroid hormone progesterone. Side chains that are structurally conserved are indicated in red; side chains that differ between the hormones are in black.

**Figure 3 ijms-21-01743-f003:**
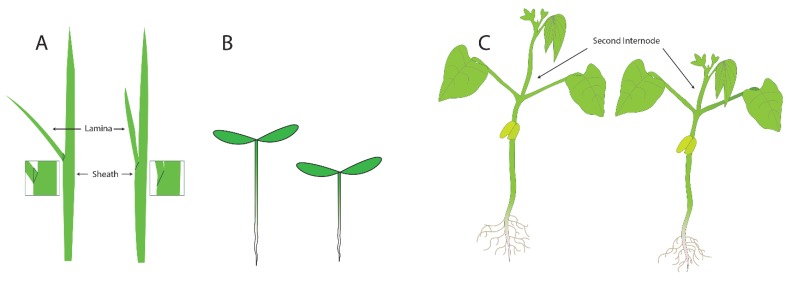
Three assays that have been used to measure the activity of brassinosteroids: (**A**) the laminar joint test, (**B**) the hypocotyl elongation assay, and (**C**) the second internode assay.

**Figure 4 ijms-21-01743-f004:**
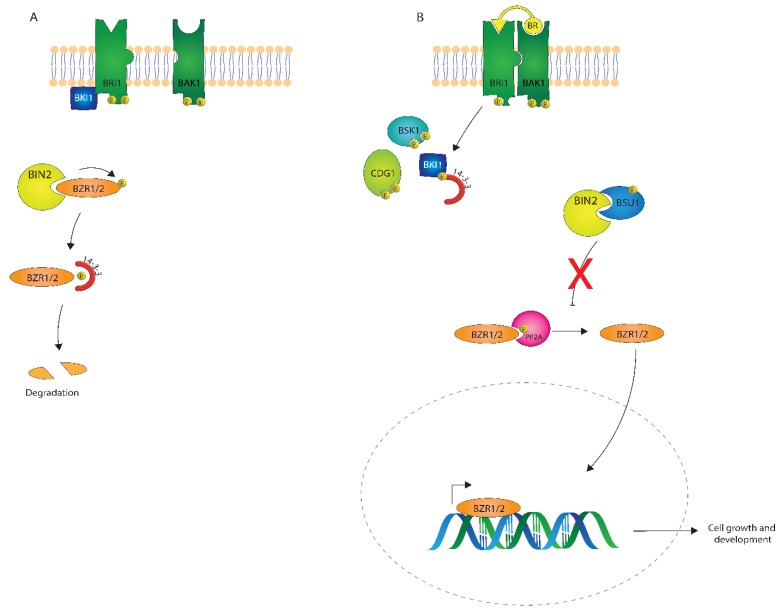
A depiction of brassinosteroid (BR) signaling both in the absence (**A**) and presence (**B**) of a BR signal.

**Figure 5 ijms-21-01743-f005:**
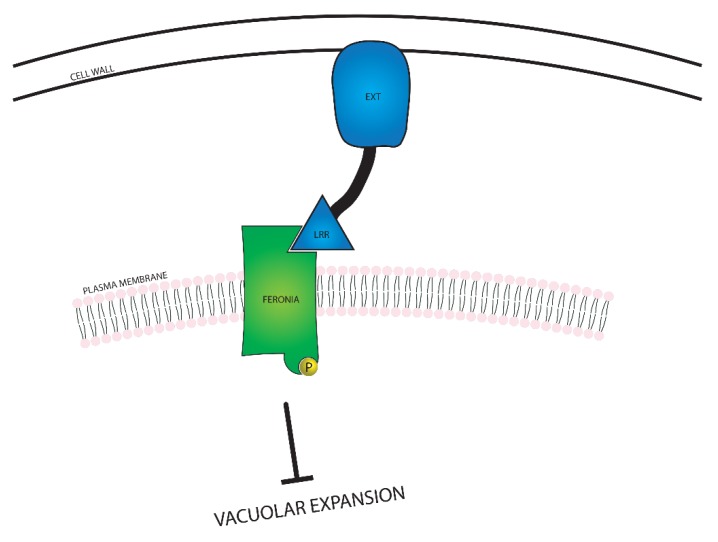
A visual representation of the relationship between FERONIA and Leucine-Rich Repeat Extensin (LRX) proteins. FERONIA resides in the plasma membrane, while the LRX protein exists in the space between the cell wall and the plasma membrane. In the event that the cell wall is close enough to the plasma membrane for the LRX protein to bind to both the cell wall and FERONIA, vacuolar expansion is inhibited.

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
