# Peer review of "The Control of Cell Expansion, Cell Division, and Vascular Development by Brassinosteroids: A Historical Perspective"

_ijms, 2020, doi:10.3390/ijms21051743_

Round 1

Reviewer 1 Report

Oh et al. presented a detailed review covering a wide range of topics related to brassinosteroids such as chemistry, physiology, molecular biology and genetics. The review started with a history of the hormone’s discovery and transition into molecular and genetics research.

A few overall suggestions:

A grammar check would help. The abstract specifically mentioned BRI1, while this is not the focus of the review. It also ended abruptly on FER. The abstract need to be reflective of the actual content of the manuscript. Too much unnecessary detail sometimes blurs he key take-home message. It is not required but maybe consider making it more succinct.

Specific comments:

Line 29: The authors mentioned BRs are considered “at the same level” as the class plant hormones. I think I understand what the authors are trying to say, but maybe can be more explicit about what “at the same level” means? Line 52: The comparison among different steroids are shown in Figure 2, but in the text the authors said Figure 1. I also am not so sure what the point is comparing the structures of BR with human hormones. Line 66-67: This might be too picky but maybe organize the text following the same order of the figure? Lamina joint test first, second interndoe last? Or figure follows text. Line 72: Why is the Steffens in red? Line 77: Lamina and sheath are parts of a leaf, I find it inaccurate saying the “cells” are called the lamina. Line 91: I would not call the petioles the “stems” of the cotyledons. Line 246: I would insert the number of the “Figure X”. Line 408: I might be lost but should this section be numbered?

Author Response

A few overall suggestions:

A grammar check would help.

We checked the grammar carefully and revised.

The abstract specifically mentioned BRI1, while this is not the focus of the review. It also ended abruptly on FER. The abstract need to be reflective of the actual content of the manuscript.

We rewrote the abstract to better reflect the manuscript, taking out BRI1 and FER

Too much unnecessary detail sometimes blurs he key take-home message. It is not required but maybe consider making it more succinct.

We took out several sections of the manuscript that were too detailed.

 Specific comments:

Line 29: The authors mentioned BRs are considered “at the same level” as the class plant hormones. I think I understand what the authors are trying to say, but maybe can be more explicit about what “at the same level” means?

We explained this more in the text

Line 52: The comparison among different steroids are shown in Figure 2, but in the text the authors said Figure 1. I also am not so sure what the point is comparing the structures of BR with human hormones.

The Figure number has been corrected and we changed the figure to just have one human hormone for comparison.

Line 66-67: This might be too picky but maybe organize the text following the same order of the figure? Lamina joint test first, second interndoe last? Or figure follows text.

Thanks for pointing this out, we fixed it.

Line 72: Why is the Steffens in red?

This is corrected.

Line 77: Lamina and sheath are parts of a leaf, I find it inaccurate saying the “cells” are called the lamina.

We have clarified this

Line 91: I would not call the petioles the “stems” of the cotyledons.

We have removed this

Line 246: I would insert the number of the “Figure X”.

Corrected, thanks.

Line 408: I might be lost but should this section be numbered? 

Corrected, thanks

Reviewer 2 Report

Article : Control of Cell Expansion, Cell Division and Vascular Development by Brassinosteroids

Review comments:

First I recommend to the author to adjust the title. The actual form of the manuscript does not comply with instructions for authors. THIS IS THE FIRST RULE FOR SUBMISSION TO ANY JOURNAL! Please write the paper according to Instructions for authors. I suggest the author spend some time to polish the manuscript and to carefully correct grammatical mistakes- should be thoroughly checked for language errors. Elements of scientific novelty should be presented in a more detailed and convincing manner. Replace the images because of their poor quality. At the conclusion section only one sentence???

Author Response

Review comments:

First I recommend to the author to adjust the title.

Thanks, we have changed the title.

The actual form of the manuscript does not comply with instructions for authors. THIS IS THE FIRST RULE FOR SUBMISSION TO ANY JOURNAL! Please write the paper according to Instructions for authors.

We have followed the instructions as best as we could. We hope it is easier to read.

I suggest the author spend some time to polish the manuscript and to carefully correct grammatical mistakes- should be thoroughly checked for language errors.

We have polished the manuscript and improved the grammar

Elements of scientific novelty should be presented in a more detailed and convincing manner.

We have worked hard to tighten the language.

Replace the images because of their poor quality.

We have replaced the images so they have better resolution

At the conclusion section only one sentence???

We concluded with the evolutionary section instead.

Round 2

Reviewer 1 Report

All reviewer questions were addressed in this revision. I thank the authors

Reviewer 2 Report

I have no comments on the manuscript except for grammatical mistakes. There are some grammatical mistakes (different theme fonts, a lot of spaces, etc.), which should be carefully corrected.